# Polycyclic Aromatic Hydrocarbons in Indoor Dust Collected during the COVID-19 Pandemic Lockdown in Saudi Arabia: Status, Sources and Human Health Risks

**DOI:** 10.3390/ijerph18052743

**Published:** 2021-03-08

**Authors:** Sultan Hassan Alamri, Nadeem Ali, Hussain Mohammed Salem Ali Albar, Muhammad Imtiaz Rashid, Nisreen Rajeh, Majdy Mohammed Ali Qutub, Govindan Malarvannan

**Affiliations:** 1Department of Family Medicine, Medical College, King Abdulaziz University, Jeddah 21589, Saudi Arabia; shalamri1@kau.edu.sa (S.H.A.); mmqutub@kau.edu.sa (M.M.A.Q.); 2Centre of Excellence in Environmental Studies, King Abdulaziz University, Jeddah 21589, Saudi Arabia; mimurad@kau.edu.sa; 3Department of Community Medicine, Medical College, King Abdulaziz University, Jeddah 21589, Saudi Arabia; hmalbar@kau.edu.sa; 4Anatomy Department, Medical College, King Abdul Aziz University, Jeddah 21589, Saudi Arabia; nrajeh@kau.edu.sa; 5Toxicological Centre, University of Antwerp, Universiteitsplein 1, 2610 Wilrijk, Belgium

**Keywords:** PAHs, COVID-19 lockdown, indoor dust, Saudi Arabia

## Abstract

To control the spread of coronavirus disease (COVID-19), Saudi Arabia’s government imposed a strict lockdown during March–July 2020. As a result, the public was confined to indoors, and most of their daily activities were happening in their indoor places, which might have resulted in lower indoor environment quality. Polycyclic aromatic hydrocarbons (PAHs) were analyzed in household dust (*n* = 40) collected from different residential districts of Jeddah, Saudi Arabia, during the lockdown period. PAHs’ levels were two folds higher than the previously reported PAHs in indoor dust from this region. We detected low molecular weight (LMW) with two to four aromatic ring PAHs in all the samples with a significant contribution from Phenanthrene (Phe), present at an average concentration of 1590 ng/g of dust. Although high molecular weight (HMW) (5–6 aromatic ring) PAHs were detected at lower concentrations than LMW PAHs, however, they contributed >90% in the carcinogenic index of PAHs. The estimated daily intake (EDI) of specific PAHs was above the reference dose (RfD) for young children in high-end exposure and the calculated Incremental Lifetime Cancer Risk (ILCR) was >1.00 × 10^−4^ for both Saudi adults and young children. The study highlighted that indoor pollution has increased significantly during lockdown due to the increased indoor activities and inversely affect human health. This study also warrants to conduct more studies involving different chemicals to understand the indoor environment quality during strict lockdown conditions.

## 1. Introduction

Polycyclic aromatic hydrocarbons (PAHs) are pervasive organic compounds mainly consisting of hydrogen and carbon atoms structurally aligned in more than one aromatic rings. They constitute a group of semi-volatile organic compounds (SVOCs) produced during incomplete combustion, mainly under high moisture, insubstantial temperature and suboptimal oxygen content conditions [1,2,3]. They are ubiquitous in the environment as a result of natural and anthropogenic activities, including organic matter pyrolysis, fossil fuels utilization, industrial and biological activities [1,2,3]. Incense burning, cooking, smoking and heating activities are primary PAH sources indoors [2,3,4]. Bakhoor is the name given to the practice of burning incense made of wood chips soaked in perfume oil, commonly practiced indoor for aesthetic reasons in the Gulf countries. According to recent studies, this might be one of the causes of indoor PAH pollution [5,6,7]. The other primary sources of PAHs and environmental pollutants in the indoor environments are shoes dust/soil and infiltrating air entered indoors during cross-ventilation [8,9,10]. Like other SVOCs these PAHs can partition among airborne particles, vapour phase, settled and surface dust [11,12].

Low molecular weight (2–4 ring) PAHs are more volatile and exist in the gaseous phase, while on the other hand, high molecular weight (4–6 ring) PAHs show inconsequential vaporization, so they are mainly found in the particulate state [12]. According to Kuo et al. [12], PAH concentrations in particulate matter significantly correlated with the quantity of dust present in the indoor air. On the other hand, settled dust, particularly that embedded in carpeted floors could serve as a reservoir for PAHs and many other pollutants in the indoor environment [10,13]. Most indoor pollutants remain indoors for the long run due to limited sunlight and ventilation availability. Consequently, these pollutants do not degrade or break down into smaller and less harmful compounds [10,14]. This study indicates that indoor dust is an indoor pollution reservoir, and its analysis can provide reliable evidence of the scale of indoor contamination [10]. Thus, the fingerprinting of indoor chemicals is also essential for human health because of the amount of time spent indoors in modern times. Several studies have reported that exposure to PAHs can cause different health problems, e.g., endocrine disruption, reproductive system abnormalities, developmental disorders, neurological disorders, skin allergies, asthma and premature births [15,16,17]. Some PAHs are well known for their carcinogenic, mutagenic, and teratogenic properties [15,17].

The outbreak of the novel SARS-CoV-2 coronavirus has caused worldwide social and economic disruptions. Within a month of its origin, on 30 January 2020, a “public health emergency of international concern” was declared by the WHO due to its high human-to-human transmission frequency and mortality and unprecedented worldwide spread [18]. Consequently, this pandemic resulted in a tremendous socio-economic disruption worldwide and resulted in a few lifestyle changes [18]. According to the best current knowledge, the most effective disease spread preventive measure against the COVID-19 pandemic was reduced human interaction with each other [19]. Consequently, most of the affected countries, e.g., EU countries, the USA, and Saudi Arabia, announced lockdowns and placed strict restrictions on people’s movement as a tool to curb the spread of COVID-19 [20]. Many studies showed the positive impact of such measures on the outdoor environment quality [18]. However, little focus is given to the environmental quality indoors where most of the public was confined during this period. In households, especially those with young children, the indoor environment might have significantly affected their playing activities due to indoor cooking and limited/lack of cross-ventilation. Therefore, it was essential to monitor indoor pollution during the curfew period and understand the dynamics of indoor chemical pollution and its impact on public health.

Indoor dust has gained importance in recent years as an exposure pathway to environmental pollutants [10,21]. There is a general shortage of information on exposure to environmental contaminants in indoor environments during lockdown situations, therefore considering this existing knowledge gap, the current study aimed to determine the occurrence of PAHs in indoor dust of Saudi households during the COVID-19 spread and lockdown period. The project’s findings represent another contribution to our knowledge about the impact of COVID-19 on the environment and health of the people.

## 2. Materials and Methods

### 2.1. Chemicals and Solvents

In this study we analyzed the occurrence of the following EPA priority list PAHs in household dust samples; acenaphthylene (Ace), anthracene (Ant), benz[a]anthracene (BaA), benzo[a]pyrene (BaP), benzo[b]fluoranthene (BbF), benzo[g,h,i]perylene (BghiP), benzo[k]fluoranthene (BkF), chrysene (Chr), dibenz[a,h] anthracene (DahA), fluorene (Flu), indeno(1,2,3-cd)pyrene (IcdP), phenanthrene (Phe), and pyrene (Pyr). Following internal standards (ISs) acenaphthylene-D10 (Ace-D10), phenanthrene-D10 (Phe-D10), and chrysene-D12 (Chr-D12) were used for the quantitative analysis. Analytical standards with >99% purity were purchased from Sigma Aldrich (Bellefonte, Pennsylvania (PA), USA). All stock solutions for the analytical standards were prepared in iso-octane, and toluene of different concentrations ranging between 0.1 and 10 µg/mL. For all internal standards, 5 µg/mL were prepared in iso-octane. Silica BondElut (500 mg, 3 mL) cartridges, acetone, dichloromethane (DCM), *n*-hexane (*n*-Hex), and *iso*-octane were of analytical grade obtained from Sigma Aldrich. All glassware used in the sample preparation were baked at 400 °C overnight and kept at 100 °C until use.

### 2.2. Sampling

For this study, indoor dust samples were collected from various households (*n* = 40) of Jeddah, Saudi Arabia, during the COVID-19 related lockdown down period (April–July 2020). Movement was strictly controlled and without work-related official permission, one could not travel outside the residential area where one lived. At the same time, people were not comfortable with meeting others due to the pandemic situation. Therefore, it was not easy to conduct a large-scale sampling campaign for the study sample. A sampling method and a questionnaire were prepared with various information such as social-economic status, area, number, and age of people sharing the household, cooking methods, dusting habits from the participating homes for the volunteers who participated in the study. In this study, families with a minimum of three people were selected, preferably with kids, and those who participated in these numbers varied from three to eight. We contacted some people with a scientific background such as our university colleagues and research students from different residential areas of Jeddah to collect household dust samples from their homes and some of their relative households for the study. Household dust was obtained from the vacuum cleaners of the respective households. Participants were asked to collect the dust from the top of their vacuum cleaner bag to have a fresh dust sample. The dust samples were wrapped in aluminum foil and sealed in individually marked zipped bags. The samples were kept in the freezer and later transferred to the lab for analysis. Each dust sample was sieved using a 200 µm mesh and samples were stored at 20 °C until analysis.

### 2.3. Sample Preparation and Quantitative Analysis

A detailed description of the sample preparation procedure is provided by Ali et al. [7]. Briefly, an accurately measured dust sample (~50 mg) was taken. After spiking with ISs, a solvent mixture (hexane/acetone (4/1, *v/v*)) was added and kept overnight to achieve equilibrium. The next day, samples were extracted using ultrasonication (20 min) followed by centrifugation (3000 rpm for 10 min). The supernatant was collected in a clean tube; the same extraction procedure was repeated twice with the leftover sediments. The extracts were pooled and brought to incipient dryness using a gentle stream of nitrogen. After drying, samples were resolubilized again in 1 mL solvent mixture (hexane and acetone) and cleaned further using BondElut silica (500 mg, 3 mL) and 10 mL solvent mixture (hexane/dichloromethane 2:1 *v/v*) for quantitative analysis. After elution, the obtained fraction was concentrated to incipient dryness under a gentle stream of nitrogen. It then was resolubilized in 100 µL of *iso*-octane for gas chromatography-mass spectrometry (GCMS) analysis. A detailed description of the instrument used for the analysis is provided elsewhere [7]. Briefly, a Shimadzu GCMS-QP2010 system was used in selected ion monitoring (SIM) mode for quantitative analysis of PAHs. A fused silica capillary column (TR5 30 M × 0.25 mm × 0.25 µm) used for the separation with the injector and ion source temperature were set at 230 °C and 280 °C, respectively. Helium was used as the carrier gas at 1.5 mL/min. The oven temperature was raised from 100 °C to 300 °C using a ramp of 8 °C/min and held for 10 min. For quantitative analysis, *m/z* 152, 162, 178, 188, 202, 228, 240, 252, 276, and 278 were used. A ten-point calibration (ranged 0.001 to 10 µg/mL) was used for the quantification with a regression coefficient greater than 0.995 for all compounds.

All the glassware used were baked at 400 °C overnight and kept at 100 °C till use. All the solvents and analytical standards used for the study were of analytical grade with high purity. Standard reference material (SRM) 2585 from the National Institute of Standards & Technology (NIST), procedural blanks (one for every eight samples), washed Na_2_SO_4_ (dust replica) spiked with a known concentration of standards were used to evaluate the procedure accuracy. The analytes’ levels found in procedural blanks corrected from the concentrations of the analysts in the samples. The values of PAHs in SRM 2585 were similar (RSD < 25%) with other reported values [22] and other studies mentioned in Appendix A. Recovery of PAHs in spiked Na_2_SO_4_ ranged between 70–125%. Recoveries were slightly higher than 110% for few PAHs, suggesting a poor connection between them and used internal standards since we used only three labelled internal standards for 13 PAHs. Simultaneously, matrices effects could also influence their recoveries, Na_2_SO_4_ was used as a surrogate for the dust in the spiked samples.

### 2.4. Human Risk Assessment Calculations

Health risk assessment of local population was calculated by per day exposure, hazard quotient (HQ), hazardous index (HI), and incremental lifetime cancer risk (ILCR). We used the following Equations (1)–(3) [23] to calculate non-carcinogenic chronic daily intake through dust ingestion, inhalation, and dermal contact. For HQ calculation of each exposure route Equation (4) was used, and calculation of HI was carried out by combining the HQ of different exposure routes (Equation (5) [23].
(1)$${\mathrm{CDI}}_{\;\mathrm{Ingestion-nca}}=C_{n}\frac{R_{\mathrm{ing}}\times \mathrm{EF}\times \mathrm{ED}}{\mathrm{BW}\times {\mathrm{AT}}_{\mathrm{nca}}}\times \mathrm{CF}$$
(2)$${\mathrm{CDI}}_{\;\mathrm{Inhalation}-\mathrm{nca}}=C_{n}\times \frac{R_{\mathrm{inh}}\times \mathrm{EF}\times \mathrm{ET}\times \mathrm{ED}}{\mathrm{PEF}\times \mathrm{BW}\times {\mathrm{AT}}_{\mathrm{nca}}}$$
(3)$${\mathrm{CDI}}_{\;\mathrm{Dermal}\;\mathrm{contact}-\mathrm{nca}}=C_{n}\times \frac{\mathrm{SA}\times \mathrm{SL}\times \mathrm{ABSd}\times \mathrm{EF}\times \mathrm{ED}}{\mathrm{BW}\times {\mathrm{AT}}_{\mathrm{nca}}}\times \mathrm{CF}$$

HQ = CDI-nca/RfD (for each exposure route)
(4)

HI = (HQ _ingestion_ + HQ _inhalation_ + HQ _dermal contact_)
(5)

In the above equations, C_n_ signifies the PAHs concentration (µg/g) in indoor dust, and for these calculations, we used 90th percentile of the concentrations. Table 1 explains the parameters of equations (1)–(3). We assumed high dust intake by adults and children, due to the prevailing dry arid and dusty conditions in Saudi Arabia throughout the year. Indoors, air conditioning is widely used by the Saudi public for cooling purposes throughout the year, which results in regular air circulation indoors and thus leads to accumulation of a high quantity of fine indoor dust particles [24].

Equations (6)–(8) were used to estimate carcinogenic risk exposure via different exposure routes. Moreover, the total carcinogenic risk was estimated by calculating the combination of all exposure routes and cancer slope factor (SF) in Equation (9) [23].
(6)$${\mathrm{CDI}}_{\;\mathrm{Ingestion}-\mathrm{ca}}=C_{n}\times \frac{\mathrm{IR}\times \mathrm{EF}}{\mathrm{ATca}}\times \mathrm{CF}$$
(7)$${\mathrm{CDI}}_{\;\mathrm{Inhalation}-\mathrm{ca}}=C_{n}\times \frac{\mathrm{EF}\times \mathrm{ET}\times \mathrm{ED}}{\mathrm{PEF}\times 24\times {\mathrm{AT}}_{\mathrm{ca}}}\times 10^{3}$$
(8)$${\mathrm{CDI}}_{\;\mathrm{Dermal}\;\mathrm{contact}-\mathrm{ca}}=C_{n}\times \frac{\mathrm{ABSd}\times \mathrm{EF}\times {\mathrm{DFS}}_{\mathrm{adj}}}{{\mathrm{AT}}_{\mathrm{ca}}}\times \mathrm{CF}$$
(9)$$\mathrm{ILRC}=({\mathrm{CDI}}_{\;\mathrm{ingestion}-\mathrm{ca}}\times \mathrm{SF}\;\mathrm{oral})+({\mathrm{CDI}}_{\;\mathrm{inhalation}-\mathrm{ca}}\times \mathrm{SF}\;\mathrm{inhalation})+({\mathrm{CDI}}_{\;\mathrm{dermal}\;\mathrm{contact}-\mathrm{ca}}\times \mathrm{SF}\;\mathrm{dermal})$$

Cancer slope factor (SF) (mg kg^−1^ day^−1^) were available for oral (7.3), dermal (25) and inhalation (3.85) routes [24]. These values were used to calculate ILRC for PAHs, BaP, and BaPE.

Estimated daily intake (EDI) was calculated via dust ingestion and air inhalation using the following Equation (10):Estimated daily intake (ng per kg BW per day) = (C_n_ × I_R_/BW) × F _time_(10)
where C_n_ indicates the concentrations of chemicals in the dust (ng/g), I_R_ is the dust ingestion rate (adults (20 (low) and100 (high)) and young children (50 (low) and 200 (high)) mg/day) and F _time_ is the fraction of time people spend in households (24 hr for both children and adults). With the lack of knowledge on these chemicals’ bioaccessibility, we assumed 100% bioaccessibility for the EDI. For these calculations, bodyweight of 70 kg for adults and 15 kg of young children was considered [7].

## 3. Results and Discussion

### 3.1. Profiling of PAHs in COVID-19 Lockdown Household Dust

In the current study, thirteen PAHs were analyzed in the collected household dust with Flu, Phe, Ant, Chr and Pyr present in all samples while other PAHs occurred with varied detection frequencies. The levels of individual PAHs are presented in Table 2. Phe was the major compound with a median value (ng/g) of 950, while Pyr (600), BkF (435), Chr (380), and Flu (255) were the other major compounds found in these samples. Among other PAHs, Ace was found in >50% samples with an average concentration of 80 ng/g, and this is understandable because of the volatile nature of Ace. The PAHs profile was dominated by low molecular weight (LMW, 2–4 ring) PAHs, as shown in the Figure 1A. Based on average concentration, Phe contributed 1/4th of total PAHs load in the dust samples while Ace contributed the least by 1% in the PAHs profile. Pyr (14%) and Chr (11%) were the other major contributors. Overall LMW (2–4 ring) PAHs contributed an overwhelming 68% in the PAHs load by average concentrations while high molecular weight (HMW) PAHs contributed the remaining 32%. The contribution of HMW PAHs was more evenly distributed with BkF (9%) dominated followed by BbF (8%), BaP (4%), IcdP (4%), BghiP (4%), and DahA (3%) (Figure 1A). This showed HMW PAHs were evenly present in the dust samples. Our present findings on PAHs profile in these dust samples were similar to those reported in literature where 3–4 aromatic ring PAHs were the major contributors in the dust [6,7,27]. Usually, LMW PAHs, due to their greater volatility, are reported in high concentrations in air samples. Contrarily, HMW PAHs are reported to be more toxic and persistent and mostly reported in settled dust [7]. Consequently, settled dust is an important source of both HMW and LMW PAHs via dust ingestion, inhalation, and dermal contact [6,7,27].

Many studies have reported the occurrence of PAHs in household dust. Average levels from the present study were lower than those found in Canada (Ottawa) [29], USA (Texas) [30] and China [31], but similar to those of Hong Kong [32] and Italy (Palermo) [33] (Figure 2). The difference in PAHs levels might be due to the difference in the automobile traffic densities, the use of coal tar in road and car parking surfaces, geographical location, ambient metrological parameters, heating sources, and regulations regarding social restrictions and indoor smoking. A number of these reasons were hinted as a proxy index for PAHs presence in the ambient environment [34]. The present study average PAH levels were higher than those reported previously from Saudi Arabia and Kuwait [6,7]. Many of the household dust samples in the present and a previous study [9,10] were collected from the same residential districts, indicating the impact of indoor activities during lockdown on PAHs from home dust. Even though the traffic was reduced on the roads due to lockdown, there was still heavy traffic during the flexible hours due to goods transportation and work-related traffic. During the lockdown, most households reported increased incense use and cooking activities (two or more-fold higher) that might increase PAHs indoors during the lockdown period (Appendix A).

### 3.2. Source Apportionment

There are various emission sources of PAHs into the ambient environment, with both pyrogenic and petrogenic processes are reported as the major sources [6,7,35]. Different relative distributional and diagnostic indexes using certain PAHs are applied to know their possible emission points [33,35,36]. Usually, LMW PAHs are reported to have petrogenic sources. Simultaneously, HMW PAHs are mainly released from pyrogenic processes, e.g., combustion of natural fossil fuels such as coal, gasoline, diesel, and natural gas [36,37]. However, many factors play a role and alter these diagnostic ratios such as adsorption of PAHs onto dust, transportation of the PAHs in the gaseous phase, and photolytic breakdown of PAH compounds [33,35]. Nonetheless, these diagnostic ratios still indicate the most likely release points of PAHs. The diagnostic ratios between Phe/Ant ranged from 0.6–15.5 with a mean value of 4.8 and Ant/(Ant + Phe) ranged between 0.06–0.62 with a mean value of 0.22. These values of Phe/Ant ratio > 9 indicate petroleum emission while a value < 9 suggests wood combustion, diesel, and gasoline cars [35]. Similarly, the Ant/(Ant + Phe) diagnostic ratio < 0.1 indicates petroleum and > 0.1 reflects combustion as the major sources [38,39]. The HMW PAHs not consistently detected in these dust samples; therefore, in the present study we omitted using the diagnostic ratios to know the possible emission sources.

The diagnostic ratio values for BaA/(BaA + Chr) varied from 0.01 to 0.75 with an average value of 0.33. Various studies have reported that a diagnostic ratio of BaA/(BaA + Chr) < 0.20 indicates a petroleum source [35], while >0.2 hints at diesel and gasoline sources, thus demonstrating both these sources for the presence of these compounds in the analyzed dust samples [35]. The Flu/(Flu + Pyr) ratio ranged between 0.06–0.68 with an average value of 0.33. Younker et al. [35] suggested that a value between 0.4 and 0.5 of Flu/(Flu + Pyr) indicates emissions from gasoline and fuel oil combustion while >0.50 signals coal and wood combustion. These 3–4 ring PAHs’ diagnostic ratios point out their carbon emissions from pyrogenic and petrogenic sources. Although we collected information on the questionnaire such as indoor smoking, cooking style, cross ventilation with outdoors, etc. to understand the PAHs’ sources in the present study. However, no clear answers were elucidated from the collected information since most of the sampled household cooking styles were similar (2–3 times, frying, boiling, baking etc.). Also, except for a couple of households, the others did not report indoor smoking. These preliminary results are based on a small set of samples. Other factors that might have affected these ratios are not studied in the present study. It was not possible to correlate the emission sources in the present study with available information from the households. Therefore, multiple emission sources such as increased incense burning increased in cooking during the lockdown, tracked in the dust during cross-ventilation, etc. all contribute significantly to PAHs indoors. To understand these PAHs sources, large scale studies are warranted with more precise details attached to the samples.

### 3.3. Human Health Risk Assessment to PAHs via Dust Exposure

Many sources produce PAHs in the ambient environment; consequently, humans are exposed to these compounds due to their ubiquity. Humans are getting exposed to PAHs via contaminated dust, food, and air. Chronic exposure to PAHs is linked with some health issues; notably, various studies have found a significantly positive correlation between lung cancer and PAH exposure. Subsequently, various regulatory agencies like the National Occupational Safety and Health Administration (OSHA), United States Agency for Toxic Substances and Disease Registry (ATSDR), the International Agency for Research on Cancer (IARC), USEPA, and the Department of Health and Human Services (DHHS) have placed them into different categories of carcinogenic compounds.

Benzo (a) pyrene equivalent carcinogenic power (BaPE) is a one way of understanding the carcinogenic power of PAHs using main toxic PAHs in equation suggested by Cecinato [40]:(11)BaPE=0.06×BaA+0.07×B(b+k)F+BaP+0.6×DahA+0.08×IcdP

BaP is the essential potent carcinogenic, and mutagenic indicator of PAHs and world health organization (WHO) has noted it as an index for the PAHs carcinogenicity. Studies have shown that the concentration of BaP has a significant positive correlation with total PAHs in both gaseous and solid phase [28,41]. Nonetheless, BaP can also be quickly decomposed by light and when exposed to oxidants in the ambient environment. The PAHs’ carcinogenicity based on alone BaP could be miscalculated, especially when other ambient conditions are not well determined [28,41]. Thus, to understand the scale of carcinogenicity risk of total PAHs, BaPE is calculated using the above equation. Knowing the contribution of other important PAHs in the carcinogenicity of BaPE/BaP was calculated, which ranged between 1.2–21.2. This value indicates a significant contribution of other important PAHs in the carcinogenic index of PAHs.

To have further information at the contribution of specific PAHs in the carcinogenic index, BaP_eq_-TEQ was calculated using the following equation:(12)$$\mathrm{BaPeq}\;\mathrm{as}\;\mathrm{TEQ}=\Sigma C_{n}\times \mathrm{Toxic}\;\mathrm{Equivalency}\;\mathrm{Factor}\;(\mathrm{TEF})$$

In the above Equation (12), C_n_ represents the concentration (ng/g) of specific PAH in the analyzed dust sample, while TEF (ng/g) is the toxic equivalence factor of individual PAHs [28]. This equation helps understanding the potential risk posed by specific PAHs. Table 2 and Figure 1B present the values and contribution in the profile of BaPeq as TEQ for individual PAHs. Although BaP and DahA were not among the significant PAHs with a cumulative contribution of only 7% in the PAHs profile (Figure 1A), these two crucial PAHs contributed up to 69% BaPeq as TEQ (Figure 1B). Unlike the simple PAHs profile (Figure 1A), HMW PAHs contributed an overwhelming 95% to BaPeq as TEQ profile, signifying the importance of these chemicals in the settled dust. Different exposure parameters were calculated for various exposure routes and using the other Equations (1)–(10) as reported above in the methodology section to investigate the health risks associated with the long term and daily exposure to PAHs via dust. As discussed above, many of the studied chemicals are reported to be carcinogenic; therefore, the main interest in calculating the ILCR was to look at the potential long-term cancer risk to the Saudi young and adult people via indoor dust exposure.

Concurrently, to calculate non-carcinogenic risk HQ and HI were calculated using Equations (1)–(5). A value of >1 for HI indicates a non-carcinogenic risk to the population [7,41]. However, the HI for PAHs with known reference dose (RfD) values was <1, which suggested low non-carcinogenic risk for the exposed public from these specific PAHs. The ILRC was calculated using Equations (6)–(9). The probabilistic ILCR assessment was highest via dust ingestion followed by inhalation and dermal route (Table 3). The USEPA recommended that the safe limit for long term cancer risk is <1.00 × 10^−4^ [7,41]. However, for total PAHs, the estimated ILCR was above the USEPA recommendation for both adults and young children, which indicates a risk to the exposed Saudi population from PAHs via dust exposure to them from indoors in current scenarios. However, we need to caution here since ILCR is calculated for long-term exposure. These samples were collected during unprecedented times, and as previous studies [6,7] indicated that the levels of PAHs in the present study are two-fold higher. However, these calculations are still significant as a primary source of PAHs i.e., industrial activities and road traffic were low. Therefore, large-scale temporal monitoring of these chemicals in our indoor and outdoor ambient environment is warranted to understand PAHs’ health risk.

For estimated daily intake (EDI), Equation (10) was used. Based on low and high dust intake, different exposure scenarios calculated using average and 90th percentile concentrations of PAHs in the dust. For most PAHs, the estimated exposure was below RfD values except for Phe and Pyr’s high-end exposure for young children (Table 4). This represents a risk for the health of young children from PAHs in the dust. However, we need cautioning that these preliminary estimates are based on small data set. Therefore, this study has its limitations, yet it indicates the likely range of PAHs exposure to the population during this lockdown period.

## 4. Conclusions

This is the first study reporting PAH levels in household dust, an indicator of indoor household pollution, collected during the COVID-19 lockdown. The ∑PAHs concentration in household dust was two times higher than the previously reported value from this region, suggesting an increase in the indoor chemical pollution during the lockdown period. LMW PAHs dominated the total PAHs profile while HMW PAHs overwhelmingly contributed to the BaPeq as TEQ profile. However, long term non-carcinogenic risk was minimal for both the young and adult population, although the estimated probabilistic ILCR was > 1.00E-4, highlighting the exposed population’s risk through dust exposure. The EDI and ILCR calculations showed a cause of concern for the susceptible population from exposure to PAHs via household dust. Dust ingestion was the primary exposure route for both adults and young children for PAH loads. Many studies have focused on the impact of the COVID-19 lockdown on the outdoor environment, but little focus is given on indoor pollution. Therefore, this study highlights another aspect of COVID-19 lockdown impact on people, which has not been studied yet. This study also extends the body of vital evidence that indoor pollution from another chemical might have increased during this period. Therefore, more studies on indoor pollution are needed to understand indoor chemical pollution dynamics during the lockdown periods and public health.

## Figures and Tables

**Figure 1 ijerph-18-02743-f001:**
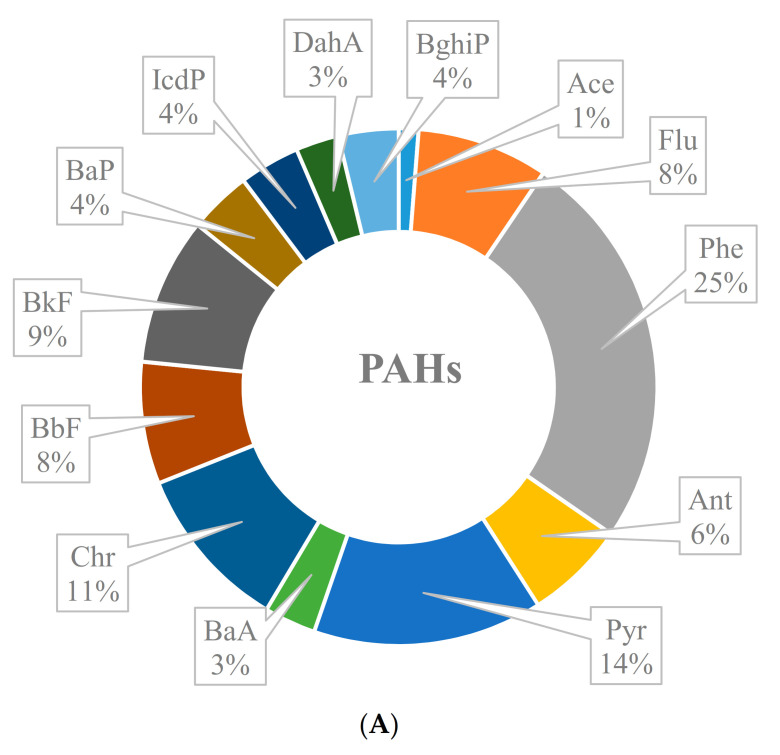
Profile of studied PAHs (**A**), and calculated BaPeq as TEQ (**B**) in indoor dust samples.

**Figure 2 ijerph-18-02743-f002:**
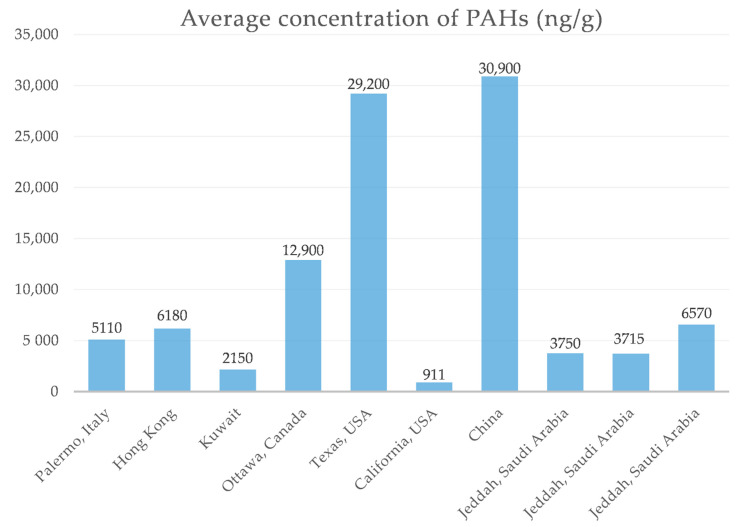
Comparison of mean concentrations (ng/g) of PAHs in indoor household dust with earlier studies from the region and other countries.

**Table 1 ijerph-18-02743-t001:** Parameters details with their acronyms used in human risk assessment equations.

Parameters	Children	Adults	Reference
Ingestion rate (R_ing_) (mg day^−1^)	200	100	[7]
Inhalation rate (R_inh_) (m^3^ day^−1^)	7.6	20	[7]
Exposure frequency (EF) (day year^−1^)	350		[25]
Duration of exposure (ED) (years)	2	30	[26]
Exposed skin area (SA) (cm^3^)	1600	6700	[26]
Dust to skin adherence factor (SL) (mg cm^−2^)	0.5		[26]
Dermal absorption factor (ABS_d_)	0.03	0.001	[25]
Particle emission factor (PEF) (m^3^ kg^−1^)	1.36 × 10^9^		[25]
Body weight (BW) (kg)	15	70	[7]
Lifetime (LT) (years)	70		[7]
Conversion factor (CF)	1 × 10^−6^		[25]
Dust dermal contact factor-age-adjusted (DFS_adj_) (mg × year kg^−1^ day^−1^)	362.4		[25]
Dust ingestion rate age-adjusted (IR) (mg × year kg^−1^ day^−1^)	113		[25]
Exposure time (ET) (hr day^−1^)	17.8	20	[26]
Average non-carcinogenic exposure time (AT_nca_)	ED × 365		[25]
Average carcinogenic exposure time (AT_ca_)	LT × 365		[25]

**Table 2 ijerph-18-02743-t002:** Concentrations (ng/g) of analyzed PAHs and BaPeq as TEQ in household dust samples collected during COVID-19 lockdown in Jeddah, Saudi Arabia.

Analytes	PAHs						BaPeq as TEQ				
	Average	StDev	Median	Mini	Max	Toxic Equivalent Factors (TEFs) [28]	Average	StDev	Median	Mini	Max
**Ace**	80	225	<LOQ	<LOQ	1130	0.001	0	0	0	0	1
**Flu**	520	1320	255	165	7710	0.001	1	1	0	0	8
**Phe**	1590	2215	950	320	12,850	0.001	2	2	1	0	13
**Ant**	400	595	215	170	3390	0.01	4	6	2	2	34
**Pyr**	910	955	600	170	4700	0.001	1	1	1	0	5
**BaA**	205	195	175	<LOQ	845	0.1	20	19	18	0	85
**Chr**	660	965	380	60	4990	0.01	7	10	4	1	50
**BbF**	485	505	300	<LOQ	2675	0.1	48	50	30	0	268
**BkF**	585	1445	435	<LOQ	7115	0.1	59	145	44	0	712
**BaP**	250	490	<LOQ	<LOQ	2155	1	250	490	2	0	2155
**IcdP**	240	260	135	<LOQ	1200	0.1	24	26	13	0	120
**DahA**	170	400	65	<LOQ	2095	1	170	400	65	0	2095
**BghiP**	235	320	90	<LOQ	1115	0.01	2	3	1	0	11
**PAHs**	6570	6700	3980	2310	37,665						
**BaPE**	385	595	100	5	2710						
**BaPE/BaP**	7.1	6.1	5.4	1.2	21.2						

**Table 3 ijerph-18-02743-t003:** The calculated potential cancer (ILCR) and non-carcinogenic (HQ and HI) risk assessment for adults and children using 90th percentile values of PAHs in household dust. * Virtually safe dose. The **Bold** values indicate values of concern.

**Chemicals**		**Adults**				**Children**			
**Non-Carcinogenic**	**RfD** [42]	**HQ-Ingestion**	**HQ-Dermal**	**HQ-Inhalation**	**HI**	**HQ-Ingestion**	**HQ-Dermal**	**HQ-Inhalation**	**HI**
Ace	0.06	1.3 × 10^−6^	8.9 × 10^−8^	7.8 × 10^−9^	1.4 × 10^−6^	2.5 × 10^−5^	3.0 × 10^−6^	1.2 × 10^−8^	2.8 × 10^−5^
Flu	0.04	6.6 × 10^−6^	4.4 × 10^−7^	3.9 × 10^−8^	7.1 × 10^−6^	1.2 × 10^−4^	1.5 × 10^−5^	6.1 × 10^−8^	1.4 × 10^−4^
Phe	0.04	4.4 × 10^−5^	2.9 × 10^−6^	2.6 × 10^−7^	4.7 × 10^−5^	8.2 × 10^−4^	9.8 × 10^−5^	4.1 × 10^−7^	9.2 × 10^−4^
Ant	0.3	1.6 × 10^−6^	1.0 × 10^−7^	9.1 × 10^−9^	1.7 × 10^−6^	2.9 × 10^−5^	3.5 × 10^−6^	1.4 × 10^−8^	3.3 × 10^−5^
Pyr	0.03	4.6 × 10^−5^	3.1 × 10^−6^	2.7 × 10^−7^	5.0 × 10^−5^	8.7 × 10^−4^	1.0 × 10^−4^	4.3 × 10^−7^	9.7 × 10^−4^
BaP	0.00014 *	3.7 × 10^−3^	2.5 × 10^−4^	2.2 × 10^−5^	4.0 × 10^−3^	6.9 × 10^−2^	8.3 × 10^−3^	3.5 × 10^−5^	7.8 × 10^−2^
**Carcinogenic**		**Ingestion dose**	**Dermal dose**	**Inhalation dose**	**ILRC**	**Ingestion dose**	**Dermal dose**	**Inhalation dose**	**ILRC (Children)**
∑PAHs		1.9 × 10^−5^	6.1 × 10^−8^	3.1 × 10^−6^	**1.5 × 10^−4^**	1.9 × 10^−5^	1.8 × 10^−6^	1.8 × 10^−7^	**1.9 × 10^−4^**
BaP		1.2 × 10^−6^	3.8 × 10^−9^	1.9 × 10^−7^	9.4 × 10^−6^	1.2 × 10^−6^	1.1 × 10^−7^	1.1 × 10^−8^	1.1 × 10^−5^
BaP × 10		1.4 × 10^−6^	4.5 × 10^−9^	2.3 × 10^−7^	1.1 × 10^−5^	1.4 × 10^−6^	1.3 × 10^−7^	1.3 × 10^−8^	1.4 × 10^−5^

**Table 4 ijerph-18-02743-t004:** Estimated daily exposure (ng/kg/bw/d) to PAHs via dust ingestion for Saudi young children and adults from their households during the lockdown period. The **Bold** values indicate values of concern.

Analytes	Adults				Toddlers			
	Exposure with Low Dust Intake (20 mg/Day)		Exposure with High Dust Intake (100 mg/Day)		Exposure with Low Dust Intake (50 mg/Day)		Exposure with High Dust Intake (200 mg/Day)	
	90th Percentile	Mean	90th Percentile	Mean	90th Percentile	Mean	90th Percentile	Mean
**Ace**	0.0	0.0	0.2	0.1	0.5	0.3	1.9	1.3
**Flu**	0.1	0.1	0.5	0.7	1.6	2.2	6.4	8.7
**Phe**	0.7	0.5	3.7	2.3	10.7	6.6	**42.7**	26.5
**Ant**	0.2	0.1	1.0	0.6	2.8	1.7	11.3	6.6
**Pyr**	0.6	0.3	2.9	1.3	8.5	3.8	**33.9**	15.1
**BaA**	0.1	0.1	0.6	0.3	1.6	0.9	6.4	3.4
**Chr**	0.5	0.2	2.4	0.9	7.0	2.7	28.1	11.0
**BbF**	0.3	0.1	1.6	0.7	4.7	2.0	18.9	8.0
**BkF**	0.3	0.2	1.4	0.8	4.1	2.4	16.6	9.7
**BaP**	0.2	0.1	1.1	0.4	3.2	1.0	12.7	4.2
**IcdP**	0.2	0.1	0.8	0.3	2.4	1.0	9.7	4.0
**DahA**	0.1	0.0	0.3	0.2	0.9	0.7	3.5	2.8
**BghiP**	0.2	0.1	0.9	0.3	2.6	1.0	10.6	4.0
**PAHs**	3.5	1.9	17.5	9.5	51.5	27.5	206	109.5
**BaPE**	0.3	0.1	1.3	0.5	3.7	1.6	14.9	6.4

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
