# Peer review of "Polycyclic Aromatic Hydrocarbons in Indoor Dust Collected during the COVID-19 Pandemic Lockdown in Saudi Arabia: Status, Sources and Human Health Risks"

_ijerph, 2021, doi:10.3390/ijerph18052743_

Round 1

Reviewer 1 Report

I reviewed the past version of this manuscript.

I have gone through the manuscript. The authors have addressed all the issues raised by me and incorporated them into the manuscript wherever required.

Therefore, I recommend the acceptance of this manuscript.

Reviewer 2 Report

The authors completely response to my question or comment. I have no more questions or comments.

This manuscript is a resubmission of an earlier submission. The following is a list of the peer review reports and author responses from that submission.

Round 1

Reviewer 1 Report

The manuscript of Sultan Hassan, A. et al. reports on polycyclic aromatic hydrocarbons in residential dust collected during COVID-19 lockdown in Saudi Arabia and estimated related cancer and non-cancer human risks. While the title of this manuscript may give readers an impression that the focus of the manuscript would be evaluating the impact of the pandemic on human health via the changes of human behaviors during pandemic - lockdown - that alters the indoor air pollution, which is an important topic to investigate, the manuscript actually did not analyze any factors that might be related with the pandemic/lockdown. The novelty of the study is moderate, as numerous studies have reported PAH composition and concentrations in residential dust. The risk assessment in this manuscript followed standard procedure and did not generate new information. Nonetheless, the PAH concentrations observed in this study still valuable and can potentially be used in further investigations.   

Major comments

  1. The authors concluded that “[t]he ΣPAHs concentrations in household dust were 2 times higher than the previously reported from this region,” which seems very interesting and makes one wonder why. It would be useful to investigate the possible determinants of the observed difference. For example, did people cook more or use incense more during the lockdown? It would be useful if data on human activities and indoor sources were collected.
  2. Although the above conclusion seems a significant finding, there is no detailed data and/or analysis were reported in the manuscript regarding this. There is only one sentence in the main text body of the manuscript, located in Line#222-223 – “The average levels of PAHs in the present study were higher than those reported previously from Saudi Arabia and Kuwait [9].” Is the PAH profile identified in this study different from previously reported? Were the samples collected from region/dwellings that were similar to where data were collected in the previous study? It would be very interesting to investigate if the two-fold difference is related to the lockdown or not.
  3. Are there any data to support an analysis to compare PAH concentrations and compositions before- and during-lockdown?
  4. The manuscript can benefit from a thorough revision from a scientific writing editor.

Specific comments

Introduction

Line#50-52, “…concentrations of PAHs in the particulate matter was significantly correlated with the quantity of dust present in the indoor air” – does this sentence mean the more dust in the indoor air, the higher PAH load in the dust?

Line#66-77, should be concise and can be greatly shortened.

Line #86-87, The sentence is a strong statement. Since this manuscript is still under review, I recommend the authors’ attention on a recently published paper - Shen, H., Shen, G., Chen, Y. et al. Increased air pollution exposure among the Chinese population during the national quarantine in 2020. Nat Hum Behav (2021). https://doi.org/10.1038/s41562-020-01018-z.

Line#88-89, not clear the meaning of the sentence - how FRs are relevant to this manuscript?   

Methods

Line#98-109, the description of sample analysis is usually provided after providing information on how samples were collected.

Line#113-115, “Since the movement was strictly controlled and it was not allowed to travel outside of the residential area. At the same time, people were not comfortable at meeting other people due to the pandemic situation” – How are these related to sampling? How did the researcher collect the samples if people were not allowed to travel outside of the residential area?

Line#115-116, it is unclear why “it was not easy to collect a large number of samples for the study” when the “participation in the study was on volunteer bases.” Isn’t participation in research projects always voluntary? How did this affect the sample size?

Line #119-120, what was the maximum number of people in the household in this study?

Line #184 “We contacted different people from different residential areas to collect a few samples from their area for the study household dust was obtained from the vacuum cleaners of the respective households” – is this one sentence or supposed to be two sentences?

Line#137-138, should read “for quantitative analysis.”

Results

Line#263-308 majority of these seem to belong to either the Methods or the Discussion sections, as it describes the calculation and/or interpretation of BaPE and ILCR.  

Reviewer 2 Report

IJERPH-1080584

The aims of the manuscript ‘Polycyclic Aromatic Hydrocarbons in Indoor Dust Collected during COVID-19 Pandemic Lockdown in Saudi Arabia: Status, Sources and Human Health Risk’ was to determine the occurrence PAHs in indoor dust of Saudi households during the COVID-19 spread and lockdown.

The manuscript is interesting, and it is a relevant contribution. However, a hypothesis is missing, and several weaknesses were found.

The citation must be reduced. No more than three citations should be used by statement.

More information regarding the preparation of stock solutions must be added.

Line 158: How a recovery of PAHs higher than 100% is explained?

Thirteen PAHs were analyzed in the collected household dust, but Eq. 11 only has six. Why?

Equations must be edited and improved. Besides, do not forget to indicate the meaning of each acronym. It is too hard to understand the Eqs. in the current form.

Please read again and carefully throughout the manuscript. It has typography mistakes, inadequate use of capital letters, and some blanks are missing.

Why Fig. 2 is added? How this Fig. contributes to the discussion. The design and discussion regarding this Fig. must be improved or throw it away.

Table and Figures captions and headings must be self-explanatory. Do not forget to indicate the meaning of each acronym and do not duplicate legends. The Figures and Tables' legends should be improved so that the main ideas and values are understandable, and the data make sense.

There are some typos and duplicated or missing words or symbols.

The arrangement and presentation of the data, besides their discussion, is not appropriate. Read and understand the manuscript in the current form is too hard.Therefore, I cannot recommend publication of this manuscript for the journal IJERPH.

Reviewer 3 Report

            This study sampled and analyzed the PAHs from home dust. It is difficult to conduct a research during COVID-19 pandemic lockdown, thus, these data is valuable. However, I really concern about sampling method and source of PAHs:

  1. the authors didn’t describe how to confirm the consistent of sampling processes from different participants, including sampling time points and spaces;
  2. in the section “source apportionment”, outdoor air seem to an important pollution source of PAHs, such as traffic emission. However, the explanation is not consistent with the description in sections “conclusion” and “abstract”. In the other words, occupants’ activities in indoors was not an important influencing factor.
  3. finally, the authors only compared the PAHs in home dust with the previous studies from other regions or countries. However, we didn’t know the really state of PAHs from home dust in the same areas before lockdown. I cannot be convinced about the effects of lockdown on PAHs from home dust.